# False memory in a second language: The importance of controlling the knowledge of word meaning

**Mar Suarez** *⊙ *ᵒ, **Maria Soledad Beato**ᵒ

Department of Basic Psychology, Psychobiology and Methodology of Behavioral Sciences, Faculty of Psychology, University of Salamanca, Salamanca, Spain

ᵒ These authors contributed equally to this work.
* marsuarez@usal.es

**Data Availability Statement:** All data files are available from the open science framework at https://osf.io/qjhs3/.

**Funding:** This research was partially supported by the University of Salamanca and Banco Santander

## Abstract

In the globalized world we live in, it is increasingly common for people to speak more than one language. Although research in psychology has been widely interested in the study of false memories with the Deese/Roediger-McDermott (DRM) paradigm, to date, there is a scarcity of studies comparing false memories in the first and the second language (L1 and L2, respectively). It is noteworthy that one of the most studied variables in the DRM paradigm, the backward associative strength (BAS), has hardly been studied in the L2. Moreover, the only study that recently examined this matter found differences in the knowledge of L2-word meaning between the high-BAS and low-BAS lists, which would hinder the interpretation of the BAS effect in L2 false memories. Taking all this into account, the current work examined false memories in the L1 (Spanish) and the L2 (English) as a function of BAS overcoming the limitations of the previous study. We selected DRM lists using both Spanish and English free association norms and lists were constructed to vary in BAS values while controlling the knowledge of word meaning. Results showed that false recognition was greater in the L1 or dominant language than in the L2 or non-dominant language. Furthermore, BAS modulated the false recognition in both the L1 and the L2. That is, false recognition was higher in high-BAS than low-BAS lists in both languages. Sensitivity index from the signal-detection theory helped us gain further insight into these results. The main findings are discussed in the light of theoretical models from both the false memory and the second language processing literature. Finally, practical implications and future research are provided.

## Introduction

The increase in bilingualism has been influenced by the fact that the world has become more globalized and the use of the Internet has been extended to most of the world's population [1]. This has meant that the possibilities for language exchange and exposure to languages other than the mother tongue have continued to grow. These circumstances have also led to an increasing number of people with a certain proficiency of a second language. Thus, it is not

through a pre-doctoral research contract attributed to MS. The funders had no role in study design, data collection and analysis, decision to publish, or preparation of the manuscript.

**Competing interests:** The authors have declared that no competing interests exist.

surprising that research in psychology has become interested in the study of the effects that learning a second language could have on cognitive functioning (for review, see [2–5]).

Furthermore, the scientific literature in psychology has also been interested in the study of false memories because of their practical implications in the field of forensic [6–9] or clinical psychology [10, 11]. The Deese/Roediger-McDermott (DRM) paradigm [12, 13] has been the most commonly used paradigm for investigating false memories in a controlled laboratory setting. In this paradigm, participants study a list of words (e.g., fountain, bridge, pool, boat, swim, fish) associated, according to free association norms [14], with a non-presented word, also called the critical lure (e.g., WATER). In a subsequent memory test, participants frequently falsely recall and/or recognized the critical lure as a previously studied word, even though it was never presented during the study phase. This is known as the false memory effect or the associative memory illusion. The robustness of the false memory effect using the DRM paradigm has been proven by studies interested in understanding the mechanisms underlying false memories [15–24] and its neural correlates [25–29], as well as studies that have found the false memory effect in clinical populations [30–34] or throughout development [35–38].

Although these two lines of research, the second language processing and the false memory, have been extensively studied separately, there is a scarcity of studies examining false memories in the first language (hereafter L1) as compared to the second language (hereafter L2). Precisely, a recent review conducted by Suarez and Beato [39] only found eight articles in the literature examining this topic. Despite the heterogeneity of the studies found in terms of language dominance, language proficiency, and age of the participants, the results of the review seem to point out that false memories occur more often in the L1 or dominant language than in the L2 or non-dominant language.

At a theoretical level, the greater false memory in the L1 than in the L2 can be accommodated by predictions of both the activation-monitoring framework (AMF) [40], a theoretical framework for the false memory effect, and the revised hierarchical model (RHM) [41, 42], a theoretical model for the processing of a second language. It is worth mentioning that these are not the only theories that have tried to explain these effects. An alternative explanation for the false memory effect is given by the fuzzy-trace theory, which states that false memory appears when the gist information of the list (matching the critical lure) is extracted, and the retrieval of verbatim representations is not enough to reject the critical lure (see [43, 44] for more details). Moreover, other theoretical models that account for the second language processing are the bilingual interactive model [45] or the ontogenesis model of L2 lexical representation [46] (see [47] for discussion). In this work we have chosen to build our theoretical foundations upon the AMF and the RHM as they both make predictions in terms of activation processes. This will allow us to bridge the language and memory literature for an integrated discussion of the results.

On the one hand, the AMF proposes that the false memory effect occurs due to the combination of two processes: activation and monitoring processes. Due to the pre-existing associations among concepts, when a DRM list is studied, the conceptual representations of those words become activated and that activation spreads to associated concepts, among which the critical lure is included. In this way, the probability of falsely identifying the critical lure as a studied word (false memory) in a posterior memory test is increased (i.e., error-inflating processes). In addition, monitoring processes can be put in place that reduce the probability of producing false memories (i.e., error-editing processes). Thus, according to the AMF, false memories occur when, first, the critical lures have been activated during the study phase and, subsequently, the monitoring processes fail preventing us from correctly retrieving the source of that activation.

On the other hand, the RHM [41, 42] explains how two languages are represented and processed in one brain, mainly focusing on cases in which the two languages have been learnt sequentially and there is a difference in language proficiency among the two (i.e., L1 = dominant language, L2 = non-dominant language). This model assumes two levels of representation, the lexical and the conceptual level, with both languages having independent lexical representations and a shared conceptual store. At the lexical level, even though words in both languages are stored independently, they are interconnected. These connections are stronger from the L2 to the L1 than in the opposite direction [48] due to the fact that the L2 was learnt by creating links between L2 words and their corresponding translation into the L1. At the conceptual level, the RHM posits that the links between words and concepts (i.e., conceptual links) are stronger in the L1 than the L2, leading to a quicker activation of the conceptual store from words in the L1 than in the L2. Because conceptual links in the L2 are weaker than in the L1, speakers with low proficiency in their second language would access concepts from words in the L2 through their L1 translation. That is, when an L2 word is encountered, it would be translated into the L1 and from that L1 lexical representation the concept would be activated. Nonetheless, once L2 learners become more proficient, the conceptual links from L2 words to the concepts become stronger.

Taken these two theoretical models together, predictions from both the AMF and the RHM can explain the greater false memories found in the dominant than the non-dominant language. According to the RHM, as speakers have stronger conceptual links in their L1 than in their L2, they will activate the conceptual representations faster and more directly from L1 words than L2 words. Thus, when participants study words in their L1, they would rapidly and automatically access the concepts and, in line with the AMF, this activation would spread throughout a well-organized network with strong connections to associatively related concepts, namely, the critical lures. This, in turn, would produce a higher level of false recognition in the L1 or dominant language than in the L2 or non-dominant language. In the case of participants who study words in their L2, they would activate the conceptual representations indirectly (i.e., through their L1 translation) or through weaker conceptual links, which would lead to the spread of activation in a less well-organized network with weaker connections to associated concepts (i.e., critical lures), thus leading to the production of fewer false memories in the L2 than in the L1.

As it was mentioned above, few studies have investigated false memories in an L2. In this regard, we found that one of the most studied variables in the DRM paradigm, the associative strength between the words of the list and the critical lure (i.e., backward associative strength or BAS), has hardly been studied in the L2. Briefly, BAS refers to the associative strength from the studied items to the critical lure [21, 22, 49]. To the best of our knowledge, only one recent study has jointly examined the effect of language dominance (i.e., L1 vs. L2) and BAS on the associative memory illusion [50]. Besides this study finding a greater false recognition in the L1 than the L2, a result consistent with the sparse previous literature; for the first time, Beato and Arndt [50] provided evidence that false recognition in the L2 was greater in high-BAS than in low-BAS lists, just as it has been reported in the L1 [40, 51–54]. However, this study has some limitations when interpreting the BAS effect in the L2. First, the most remarkable limitation is related to L2-word knowledge. Although it was the first time in the literature that special attention was drawn to the importance of assessing participants' knowledge of the meaning of the L2 words presented in the experiment; unfortunately, Beato and Arndt [50] found that there were differences in the knowledge of L2-word meaning between high-BAS and low-BAS lists. Albeit they provided evidence in their analyses that this difference did not seem to be affecting the BAS effect on the L2, it would be advisable that there were no differences in the knowledge of word meaning between the high- and low-BAS lists. Second, the

authors used DRM lists with three critical lures, which, according to Cadavid et al. [55], could be reducing the false recognition rate compared to using only one critical lure per list due to the engagement of error-editing processes. Finally, Beato and Arndt used DRM lists that varied in the levels of both backward (BAS) and forward associative strength (FAS or the associative strength from the critical lure to the studied items; see [49] for a recent study independently examining the effect of BAS and FAS on false memory). Although these authors controlled the FAS values between high- and low-BAS lists, we believe that, in order to study the effect of BAS on false memory in isolation, it would be advisable to use DRM lists with FAS values close to zero.

Given the paucity of research on this topic, the present experiment pursues the general goal of examining false memories in the L1 and the L2 as a function of the associative strength between the words of the lists and the critical lures (i.e., BAS) overcoming the limitations of Beato and Arndt's [50] study. Why is it interesting to examine the effect of BAS on false memories in the L1 and the L2? Studying this effect is of special interest because it allows us to examine both the activation of concepts from words in the L1 and the L2, and how this activation spreads throughout the semantic memory, thus contributing valuable information to the literature on the organization of the L1 and L2 lexicon [46, 56, 57].

Furthermore, why is it important to conduct this replication study and to address the methodological limitations of the previous article [50]? On the one hand, due to the replication crisis that the field of psychology has been suffering [58], and considering that reproducibility should be one of the core principles of science, replication studies are a way to improve this matter. Following the Open Science Collaboration [59], a single study provides tentative evidence, while a successful replication provides confirmatory evidence. On the other hand, addressing the methodological limitations will help us clarify and validate Beato and Arndt's findings on the effect of BAS on false memories in the L2, and this will have theoretical implications. These authors found a higher false recognition in the L1 than L2, and the BAS effect occurred not only in the L1 but also in the L2. These results go in line with the predictions of the RHM and the AMF, as it was explained above. However, a core assumption of these predictions is that participants know the meaning of all the words, otherwise they will not be able to access the conceptual store from the lexical representations and no activation would be spread to associated concepts. Therefore, as Beato and Arndt [50] provided evidence that second-language learners' knowledge of L2 stimuli was far from perfect, there is an alternative explanation for these memory effects. Precisely, a plausible explanation would be that differences in the knowledge of word meaning across languages can explain some or all language dominance effects on false memory. In the present experiment we will include a measure of the knowledge of L2-word meaning to rule out this plausible interpretation of previous findings and contribute advancing the theory.

Specifically, in this work we proposed the following objectives and hypotheses. First, regarding word knowledge, we controlled for and analyzed the knowledge of the meaning of the words included in the L2 DRM lists. This was critical because if we want to understand how the conceptual representations are accessed from words in the L2, it is essential to use words whose meaning is known, particularly important for the critical lures. If word meaning was not known, when participants study lists of words, their conceptual representations would not be activated and, therefore, activation would not spread to associated concepts (i.e., critical lure). Moreover, if the critical lures were unknown, even though the list words were known, the activation could never reach and trigger the critical lures. Consequently, the associative memory illusion could not be observed. This is especially important for second language learners who are unlikely to know the meaning of all L2 words. To this end, prior to the selection of the materials, we conducted a study on a large set of words to estimate the percentage of

knowledge of word meaning by L2 speakers. This was useful for selecting high- and low-BAS DRM lists that did not differ in their estimated knowledge of word meaning. Furthermore, at the end of the experiment, participants filled in a translation test to have an objective measure of the knowledge of L2-word meaning. We expected that there would be no significant differences in the knowledge of L2-word meaning between high- and low-BAS lists.

Second, using DRM lists with one critical lure and FAS values of virtually zero, we expected to replicate the false recognition effect in the L2. In other words, we hypothesized, in line with previous studies, that participants would falsely recognize a higher proportion of critical lures than distractors in the L2 recognition test [50, 60–65].

The third objective was to examine the effect of BAS on false recognition in the L2. Replicating the only previous study that, to our knowledge, has examined this topic [50], we expected to find a greater false recognition in the L2 for high-BAS than low-BAS lists, but this time making sure that high- and low-BAS lists do not differ in terms of the knowledge of L2-word meaning. This hypothesis goes in line with the predictions of the AMF [40] as it states that the greater the associative strength between the studied words and the critical lure (i.e., BAS), the greater the activation that the latter will receive and, therefore, the greater the probability of false recognition. In other words, the AMF predicts that the higher the BAS, the greater the probability of false memories.

The fourth and last objective was to compare the false recognition between the L1 and the L2 in both high-BAS and low-BAS lists. Predictions from both the RHM and the AMF led us to expect a higher false recognition in the L1 than the L2, both in high- and low-BAS lists. Due to the stronger conceptual links in the L1 than in the L2 [41], people would fully access conceptual representations faster and more automatically from words in the L1 compared to the L2. Based on this idea, one would expect that, during the study phase, critical lures would be activated more automatically when studying lists of words in the L1 than in the L2. This greater activation of the critical lures from words in the L1, as compared to the L2, would lead to a greater false recognition of L1 critical lures during the memory test [40]. This is precisely the result found in the few previous studies on the effect of language dominance on false recognition (for a review, see [39]).

## Method

### Participants

A total of 120 undergraduate students ($M_{age}$ = 19.70, $SD$ = 2.25, 87.50% women) voluntarily participated in this study. All participants were native Spanish speakers and were living in Spain at the time of the experiment. In addition, they had learned English as a second language (L2) in an academic context. On average, they started learning their L2 when they were 4.45 years of age ($SD$ = 1.72) and continued studying it for an average of 13.60 years ($SD$ = 2.00). Also, participants self-assessed their L2 proficiency as moderate, giving a mean score of 6.00 ($SD$ = 1.61) on a Likert-type scale ranging from 1 (elementary proficiency) to 10 (native-speaker level). Half of the participants performed the task in their L1 (i.e., Spanish) and the other half in their L2 (i.e., English). The sociodemographic characteristics of these two groups are shown in Table 1. There were no statistically significant differences between the groups in sociodemographic characteristics. The Bioethics Committee at the University of Salamanca approved this study.

Sample size was set by reviewing the number of participants included in recent (i.e., 2010–2020) previous studied published in impact journals (see [66] for a similar procedure). Specifically, we identified 65 experiments whose experimental conditions were similar to those of our work (i.e., including young healthy adults and using the standard DRM paradigm to assess

**Table 1. Participants' sociodemographic characteristics by experimental condition: L1 (Spanish) and L2 (English).**

| | Language of the task | |
| --- | --- | --- |
| | **L1 (Spanish)** | **L2 (English)** |
| *N* | 60 | 60 |
| Gender female/male | 52/8 | 53/7 |
| Years of age | 19.65 (2.86) | 19.75 (1.42) |
| L1/L2 | Spanish/English | Spanish/English |
| Age of acquisition L2 | 4.40 (1.63) | 4.49 (1.81) |
| No. years of L2 training | 13.62 (2.03) | 13.58 (1.98) |
| L1 self-assessment from 1 to10 | 10 (0.00) | 10 (0.00) |
| L2 self-assessment from 1 to 10 | 5.77 (1.52) | 6.23 (1.66) |

Standard deviations are shown in brackets.

false recognition) and observed that the mean sample size was 55.94 participants per group. For our experiment, we rounded the sample size to 60 participants per group. Furthermore, a sensitivity analysis was performed by means of G*Power 3.1 [67]. Particularly, it was conducted for the repeated measures ANOVA (within-between interaction) setting the power at .80 and alpha at .05. This analysis indicated that our sample (120 participants) had sufficient power to detect small effect sizes ($f < .10$) in our experimental design.

## Materials

Thirty-two DRM lists were selected, 16 lists in Spanish and 16 lists in English. All materials are freely available at https://osf.io/qjhs3/. Each list was composed of one critical lure and six associates through backward associative strength (BAS). In each language, eight lists were selected to have high-BAS values and the other eight were selected to have low-BAS values (see Table 2 for descriptive statistics).

On the one hand, the 16 Spanish DRM lists were obtained from a previous normative study [68]. The BAS value for each list was the average of the BAS values between each studied word and the critical lure. The DRM lists were selected so that the eight high-BAS lists had greater BAS values than the low-BAS lists, $t(7.58) = -11.21$, $p < .001$, 95% CI [-0.28, -0.18], $d = -5.60$. The high- and low-BAS lists did not differ in terms of FAS values, $t(14) = 0.37$, $p = .718$, 95% CI [-0.01, 0.02], $d = 0.18$, being these values extremely low (close to zero) for all lists. Furthermore, words from the two BAS conditions did not differ in terms of word length (i.e., number of letters), $t(14) = 1.78$, $p = .097$, 95% CI [-0.10, 1.06], $d = 0.89$, nor in their frequency (i.e., logarithmic frequency from EsPal [69]), $t(14) = 0.62$, $p = .542$, 95% CI [-0.22, 0.40], $d = 0.31$, which was, on average, 1.13 ($SD = 0.30$) for high-BAS lists and 1.22 ($SD = 0.27$) for low-BAS lists.

**Table 2. BAS and FAS values of the DRM lists as a function of backward associative strength (high-BAS and low-BAS lists) and language (L1 and L2).**

| | | L1 (Spanish) | | L2 (English) | |
| --- | --- | --- | --- | --- | --- |
| | | **High BAS** | **Low BAS** | **High BAS** | **Low BAS** |
| BAS | *M (SD)* | .28 (.06) | .05 (.01) | .28 (.06) | .05 (.01) |
| | Range | .20 –.36 | .03 –.06 | .20 –.36 | .03 –.06 |
| FAS | *M (SD)* | .01 (.01) | .01 (.01) | .01 (.01) | .01 (.01) |
| | Range | .00 –.03 | .00 –.04 | .00 –.03 | .00 –.03 |

On the other hand, the 16 English DRM lists were created from Nelson et al.'s [14] free association norms. These lists were created based on different criteria. First, concerning the knowledge of L2-word meaning, we conducted a previous study which was intended (1) to allow us to include in our experiment lists of words whose meaning was known by most of the participants, and (2) to assure that high- and low-BAS lists did not differ in the estimated knowledge of word meaning. In this previous study, 120 participants from the same population as the participants from the present experiment took part voluntarily. A pool of 768 words from the Nelson et al.'s [14] database that were suitable for creating DRM lists were selected. Each participant was asked to write down the Spanish translation of 84–150 English words. These data provided us with an estimate of the meaning knowledge for each English word. The words finally selected for our English DRM lists were known, on average, by 85.56% ($SD$ = 18.22) of the participants from this previous study. In addition, high- and low-BAS lists were selected such that there were no statistically significant differences in the estimated knowledge of word meaning (84.01% and 88.52%, respectively), $t(14)$ = 1.06, $p$ = .306, 95% CI [-4.60, 13.62], $d$ = 0.53.

The second criterion for the selection of the English DRM lists was regarding associative strength. Specifically, English DRM lists were created to be as similar as possible to the Spanish lists in terms of BAS and FAS values (see Table 2). Spanish and English DRM lists did not differ in their BAS values, $t(30)$ = -0.01, $p$ = .989, 95% CI [-0.09, 0.09], $d$ = -0.01, nor in their FAS values, $t(30)$ = -0.79, $p$ = .435, 95% CI [-0.01, 0.01], $d$ = -0.28. Also, as L1 lists, high-BAS L2 lists were found to have higher BAS values than low-BAS L2 lists, $t(7.50)$ = -11.68, $p < .001$, 95% CI [-0.28, -0.19], $d$ = -5.84. At the same time, high- and low-BAS L2 lists did not differ in their FAS values, $t(14)$ = -0.85, $p$ = .409, 95% CI [-0.01, 0.01], $d$ = -0.43, word length, $t(14)$ = 0.29, $p$ = .779, 95% CI [-0.59, 0.77], $d$ = 0.14, or frequency (i.e., logarithmic frequency based on SUBTLEX$_{US}$ [70]), $t(14)$ = 0.48, $p$ = .641, 95% CI [-0.19, 0.30], $d$ = 0.24, with frequency mean values of 1.61 ($SD$ = 0.22) and 1.66 ($SD$ = 0.24) for high- and low-BAS lists, respectively.

From the set of 16 DRM lists in each language, 10 were used in the study phase, half with high-BAS values and the other half with low-BAS values. The remaining six lists served as distractors in the recognition test (the associated words of these lists are called distractors, and the critical lure is called critical distractor). Three versions of the experimental task were created so that all lists served as both studied and distractor lists. The recognition test consisted of 80 words, of which 40 were studied words (4 per list, the first, third, fourth, and sixth associates), 10 critical lures, and 30 words from the distractor lists (24 distractors, which were the first, third, fourth, and sixth associates of each distractor list, and 6 critical distractors, the critical lure of each distractor list).

Since we were using an estimation of the knowledge of L2-word meaning, and it could differ from the actual knowledge of the participants, a translation test was created for the words included in the English lists of each version of the experimental task. Concretely, this test included the 100 English words that participants would have seen during the experiment (i.e., studied words, critical lures, distractors, and critical distractors).

## Procedure

The experiment was conducted in group sessions of up to 19 participants. Each session lasted, approximately, half an hour. Each participant was provided with a computer and a keyboard to perform the task. For this experiment, a blocking by language procedure [71] was used for which participants were randomly assigned to one of two language conditions: L1 (i.e., Spanish) and L2 (i.e., English). This blocking by language procedure allowed us to prevent the experimental context from promoting the intentional translation of L2 words into the L1 [72],

and thus, to explore the more automatic aspects of L2 processing (see [73–75] for further discussion on the importance of language mode when studying word processing in more than one language).

In addition, participants were equally distributed across one of the three versions of the experimental task in each language. Before starting the experiment, participants signed the written informed consent. Afterwards, they filled in a socio-demographic questionnaire requesting the following information: age, gender, first language, L2 age of acquisition, years of formal L2 training, and two Likert scales to self-assess their knowledge in their L1 and L2 (1 = elementary knowledge, 10 = native-speaker level). Subsequently, the experimenter read aloud the instructions for the study phase as each participant followed them on their computer screen (complete instructions in Spanish, as well as their approximate English translation, are freely available at https://osf.io/qjhs3/).

Prior to the study phase, participants were presented with a practice list to familiarize them with the format and presentation rate. Then, in the study phase, they were presented with the associates of 10 DRM lists to be studied in a single large list of 60 words blocked per list. The order of the lists was randomized for each participant. The stimulus presentation sequence in the study phase began with a fixation cross appearing in the center of the computer screen for 1,000 ms. Then, the studied words were presented visually at a constant rate (2,000 ms) in the center of the computer screen in lower case letters. A 36-point black Verdana font on a white background was used.

After the study phase, participants performed the recognition test. They were informed that they would be presented with words one at a time in the center of the computer screen. Their task was to judge whether each word had been previously presented in the study phase by pressing the corresponding key on the keyboard. Participants were instructed to place their index fingers on each key throughout the recognition test and to try to respond as quickly as possible, trying not to make mistakes. Words were presented in a random order for each participant using E-Prime 2.0 [76]. Subsequently, those participants who had performed the task in their L2 carried out a translation test of all the English words that they had seen during the experiment (studied words, critical lures, distractors, and critical distractors) using a pencil and paper questionnaire. This questionnaire included a list with 100 English words and participants had to write the Spanish translation next to each word. In case they did not know or could not guess the meaning of a word, they could leave it blank and continue with the next word. Each participant's translation test response to each word was coded by a trained researcher as 1 if the translation was correct and 0 if it was incorrect.

**Data analyses.** Analyses of variance (ANOVAs) were performed using JASP Team [77]. Across all analyses, the alpha level was set at .05, effect sizes are reported with Cohen's *d* and partial eta squared ($\eta^2_p$), and the 95% confidence intervals (CI) for the mean difference are detailed. Where appropriate, in the repeated measures ANOVAs, degrees of freedom were corrected using the Greenhouse-Geisser estimator. Bonferroni correction was used for post hoc tests and the *p*-value reported in these tests were the adjusted values for multiple comparisons.

## Results

### L2: Knowledge of word meaning

The results of the translation test, as expected, showed that participants did not know the meaning of all L2 words (see Table 3). Participants knew, on average, the meaning of 86.05% of the words, a remarkably similar percentage to that obtained in the pilot study (i.e., 85.56%). In order to examine if the knowledge of word meaning differed among the type of words, a repeated measures ANOVA was performed with the factor word type (studied words, critical

**Table 3. Mean proportion of words whose meaning was known by participants as a function of type of word and BAS.**

| Word type | High BAS | Low BAS |
|---|---|---|
| Studied words | .83 (.17) | .84 (.15) |
| Critical lures | .98 (.06) | .97 (.08) |
| Distractors | .89 (.17) | .87 (.15) |
| Critical distractors | 1.00 (.00) | .97 (.09) |

Standard deviations are shown between brackets.

lures, distractors, critical distractors) on the proportion of words whose meaning was known by participants. The ANOVA showed a main effect of word type, $F(1.56, 92.16) = 53.37$, $p < .001$, $\eta^2_p = 0.475$. Post hoc comparisons using Bonferroni correction showed that the meaning of the studied words ($M = .84$) was known to a lesser extent than the meaning of the critical lures ($M = .98$), $t(59) = -8.92$, $p < .001$, 95% CI [-0.18, -0.10], $d = -1.15$, and critical distractors ($M = .99$), $t(59) = -9.61$, $p < .001$, 95% CI [-0.19, -0.11], $d = -1.24$, both with near-perfect knowledge. Furthermore, the meaning of the distractors ($M = .85$) was also known to a lesser extent than that of the critical lures, $t(59) = 8.24$, $p < .001$, 95% CI [0.09, 0.17], $d = 1.06$, and critical distractors, $t(59) = -8.93$, $p < .001$, 95% CI [-0.18, -0.10], $d = -1.15$. However, no differences were found in the proportion of words whose meaning was known between critical lures and critical distractors, $t(59) = -0.69$, $p = 1.000$, 95% CI [-0.05, 0.03], $d = -0.09$, nor between studied words and distractors, $t(59) = -0.68$, $p = 1.000$, 95% CI [-0.05, 0.03], $d = -0.09$. In other words, the knowledge of the meaning of the critical lures of L2 DRM lists was almost perfect. In addition, the meaning of the critical lures was known more often than the meaning of the associates of the lists, regardless of whether the DRM lists served as studied or distractor lists.

A crucial check for our experiment was whether lists with high and low BAS differed in the knowledge of word meaning. To this end, we conducted a 2 (word type: studied words, critical lures) x 2 (BAS: high, low) repeated measures ANOVA on the proportion of words whose meaning was known. This analysis showed a main effect of word type, $F(1, 59) = 62.44$, $p < .001$, $\eta^2_p = 0.514$, which indicated that, as discussed in the previous analysis, critical lures' meaning was known more often than studied words' meaning. Furthermore, the main effect of BAS was not significant, $F(1, 59) = 0.05$, $p = .827$, $\eta^2_p = 0.001$, indicating that the knowledge of word meaning did not differ between high- and low-BAS conditions. Finally, a significant Word type x BAS interaction was found, $F(1, 59) = 4.05$, $p = .049$, $\eta^2_p = 0.064$. As shown in Table 3, the mean difference between high-BAS and low-BAS lists goes in the opposite direction for the studied words (i.e., the numerical value of the mean was higher for low-BAS than high-BAS lists) as compared to the critical lures (i.e., the numerical value of the mean was higher for high-BAS than low-BAS lists). However, what is crucial for the interpretation of this interaction is that the post hoc comparisons with Bonferroni correction revealed no statistically significant differences in either of these two comparisons. That is, we did not find statistically significant differences in the proportion of words whose meaning was known between high- and low-BAS lists, neither for the studied words (.83 and .84, respectively), $t(59) = -1.21$, $p = 1.000$, 95% CI [-0.04, 0.02], $d = -0.17$, nor for the critical lures (.98 and .97, respectively), $t(59) = 1.53$, $p = .773$, 95% CI [-0.01, 0.05], $d = 0.18$. These results indicate that the differences that might be found in the subsequent analyses between high- and low-BAS lists could not be due to differences in the knowledge of word meaning.

## L1 DRM illusion: The effect of associative strength

Table 4 shows the proportions of "yes" responses in the recognition test to each type of word (i.e., studied words, critical lures, distractors, and critical distractors) as a function of language (i.e., L1 and L2) and BAS (i.e., high and low). Moreover, following the signal detection theory [78], the sensitivity index $d'$ was included for the list items and the critical lures.

First, to check whether we replicated the false recognition effect in the L1, a repeated measures ANOVA was conducted with the factor word type (studied words, critical lures, distractors, critical distractors) on the proportion of "yes" responses on the recognition test. This analysis yielded a main effect of word type, $F(2.18, 129.50) = 329.71$, $p < .001$, $\eta^2_p = 0.848$. The Bonferroni post hoc comparisons indicated that the proportion of "yes" responses was higher for studied words (i.e., true recognition; $M = .78$) than for critical lures (i.e., false recognition; $M = .45$), $t(59) = 13.05$, $p < .001$, 95% CI [0.26, 0.40], $d = 1.69$, as well as for distractors (i.e., false alarms; $M = .05$), $t(59) = 28.43$, $p < .001$, 95% CI [0.66, 0.79], $d = 3.67$, and for critical distractors (i.e., critical false alarms; $M = .15$), $t(59) = 24.75$, $p < .001$, 95% CI [0.56, 0.70], $d = 3.20$. Furthermore, participants gave more "yes" responses to critical lures than to distractors, $t(59) = 15.38$, $p < .001$, 95% CI [0.32, 0.46], $d = 1.99$, and critical distractors, $t(59) = 11.70$, $p < .001$, 95% CI [0.23, 0.37], $d = 1.51$. These confirm that the L1 (Spanish) DRM lists used in this experiment produced the false memory effect. Finally, the false alarms to critical distractions were greater than to distractors in the L1, $t(59) = -3.69$, $p = .002$, 95% CI [-0.16, -0.03], $d = 3.20$.

Once we have proved that our lists produced the false memory effect in the L1, we wanted to check whether we replicated the BAS effect. A 2 (word type: studied words, critical lures) x 2 (BAS: high, low) repeated measures ANOVA showed a main effect of word type, $F(1, 59) = 127.57$, $p < .001$, $\eta^2_p = 0.684$. The proportion of "yes" responses on the recognition test was greater for the studied words than for the critical lures (.78 and .45, respectively). Moreover, a main effect of BAS was found, $F(1, 59) = 19.63$, $p < .001$, $\eta^2_p = 0.250$. There was a higher proportion of "yes" responses for words in the high-BAS lists ($M = .68$) than in the low-BAS lists ($M = .55$). Lastly, a Word type x BAS interaction was found, $F(1, 59) = 16.33$, $p < .001$, $\eta^2_p = 0.217$. The Bonferroni post hoc comparisons revealed that false recognition was higher in high- than low-BAS lists (.54 and .35, respectively), $t(59) = 5.97$, $p < .001$, 95% CI [0.10, 0.28], $d = 0.63$, whereas no statistically significant differences were found between high- and low-BAS lists in true recognition (80 and .76, respectively), $t(59) = 1.13$, $p = 1.000$, 95% CI [-0.05, 0.12], $d = 0.20$.

**Table 4. Mean proportion of "yes" responses to the different word types and the sensitivity index ($d'$) as a function of language and BAS.**

| Word type | L1 (Spanish) | | L2 (English) | |
|---|---|---|---|---|
| | **High BAS** | **Low BAS** | **High BAS** | **Low BAS** |
| Studied words | .80 (.14) | .76 (.14) | .81 (.14) | .77 (.14) |
| List-item $d'$ | 2.64 (0.75) | 2.49 (0.76) | 2.19 (0.82) | 1.97 (0.82) |
| List-item $d'$ (meaning known) | | | 2.28 (0.84) | 2.05 (0.91) |
| Critical lures | .54 (.30) | .35 (.21) | .29 (.24) | .17 (.20) |
| Critical-lure $d'$ | 1.00 (1.13) | 0.59 (0.79) | 0.43 (0.84) | -0.04 (0.77) |
| Critical-lure $d'$ (meaning known) | | | 0.43 (0.84) | -0.04 (0.79) |
| Distractors | .05 (.08) | .05 (.08) | .16 (.15) | .17 (.14) |
| Critical distractors | .17 (.21) | .12 (.18) | .11 (.18) | .14 (.19) |

Standard deviations are shown between brackets.

## L2 DRM illusion: The effect of associative strength

Parallel analyses to those in the L1 will now be discussed on the L2 data. A repeated measures ANOVA was conducted with the factor word type (studied words, critical lures, distractors, critical distractors) on the proportion of "yes" responses on the L2 versions of the recognition test. This analysis showed a main effect of word type, $F(2.59, 152.84) = 347.76$, $p < .001$, $\eta^2_p =$ 0.855, and Bonferroni post hoc comparisons indicated that the results went in line with the ones found for the L1. Precisely, the proportion of true recognition was higher than the proportion of false recognition (.79 and .23, respectively), $t(59) = 23.72$, $p < .001$, 95% CI [0.50, 0.63], $d = 3.06$, and higher than the proportion of false alarms to distractors ($M = .17$), $t(59) =$ 26.48, $p < .001$, 95% CI [0.56, 0.69], $d = 3.42$, and to critical distractors ($M = .13$), $t(59) =$ 28.15, $p < .001$, 95% CI [0.60, 0.73], $d = 3.63$. Moreover, participants gave more "yes" responses to critical lures than to distractors, $t(59) = 2.76$, $p = .038$, 95% CI [0.01, 0.13], $d = 0.36$, and critical distractors, $t(59) = 4.43$, $p < .001$, 95% CI [0.04, 0.17], $d = 0.57$. This confirms that the L2 (English) DRM lists created in this experiment produced the false memory effect in second-language learners, although apparently to a lesser extent than in the L1 (L2: .23 vs. L1: .45). Finally, no differences were found between the false alarms to the distractors and the critical distractors in the L2, $t(59) = 1.67$, $p = .584$, 95% CI [-0.02, 0.10], $d = 0.22$.

To examine the effect of BAS on true and false recognition in the L2, we executed a 2 (word type: studied words, critical lures) x 2 (BAS: high, low) repeated measures ANOVA. This analysis revealed a main effect of word type, $F(1, 59) = 385.91$, $p < .001$, $\eta^2_p = 0.867$, showing that the proportion of "yes" responses on the recognition test were greater for studied words than for critical lures (.79 and .23, respectively). A significant main effect of BAS was also found, $F(1, 59) = 17.12$, $p < .001$, $\eta^2_p = 0.225$, indicating that the proportion of "yes" responses was greater in high-BAS lists ($M = .55$) than in low-BAS lists ($M = .47$). Finally, a Word type x BAS interaction was observed, $F(1, 59) = 5.21$, $p = .026$, $\eta^2_p = 0.081$. Post hoc comparisons using Bonferroni correction indicated that there was an effect of BAS on false recognition, being the critical lures of the high-BAS lists falsely recognized more often than the critical lures of the low-BAS lists (.29 and .17, respectively), $t(59) = 4.57$, $p < .001$, 95% CI [0.05, 0.20], $d = 0.48$. However, the effect of BAS was not observed for the studied words (.81 and .77, respectively), $t(59) = 1.41$, $p = .962$, 95% CI [-0.04, 0.11], $d = 0.27$.

## L1 versus L2 DRM illusion as a function of associative strength

Once we have confirmed that in our experiment the false memory effect was found in both the L1 and the L2, we will compare the magnitude of the effect between the two languages. To compare true and false recognition in the L1 and the L2, it would not be appropriate to use the proportion of "yes" responses to the studied and critical words in each language, since participants produced more false alarms in the L2 than in the L1 (see Table 4). Therefore, to be able to compare true and false recognition more reliably in both languages, we calculated the d prime sensitivity index ($d'$) derived from the signal detection theory [78]. The $d'$ index in the DRM paradigm context allows us to obtain a measure of the amount of information encoded at study about both the studied words and the critical lures, as compared to distractors and critical distractors, respectively, which serve as controls [16, 79–83].

Briefly, $d'$ is the standardized difference between the proportion of hits and false alarms: z[P (hits)]—z[P(false alarms)]. Two $d'$ indexes were calculated in each language (see Table 4): $d'$ for studied words (hereafter, list-item $d'$) and $d'$ for critical lures (hereafter, critical-lure $d'$). On the one hand, to obtain list-item $d'$, the $z$-score of the proportion of hits to the studied words minus the $z$-score of the proportion of false alarms to distractors was calculated for each participant. On the other hand, critical-lure $d'$ was calculated by subtracting the $z$-scores of the

proportion of "yes" responses to the critical lures from the $z$-scores of the proportion of false alarms to critical distractors. It should be noted that participants' perfect accuracy (i.e., hits = 1 and false alarms = 0) was adjusted to avoid obtaining infinite $d'$ values. Specifically, proportions equal to 0 were replaced by 1/2N and proportions equal to 1 by 1–(1/2N), where N is the number of trials on which the proportion is based [84].

Using $d'$ values as the dependent variable, we conducted a 2 (word type: studied words, critical lures) x 2 (BAS: high, low) x 2 (language: L1, L2) ANOVA, with word type and BAS as within-subject variables and language as a between-subject variable. This analysis showed a significant main effect of word type, $F(1, 118) = 459.98$, $p < .001$, $\eta^2_p = 0.796$, indicating that $d'$ values were higher for the studied words than for the critical lures (2.32 and 0.50, respectively). In addition, a significant main effect of BAS was found, $F(1, 118) = 22.61$, $p < .001$, $\eta^2_p = 0.161$, with high-BAS lists yielding higher $d'$ values than low-BAS lists (1.57 and 1.25, respectively). A significant main effect of language was also found, $F(1, 118) = 37.49$, $p < .001$, $\eta^2_p = 0.241$, showing that $d'$ values were higher in the L1 than in the L2 (1.68 and 1.14, respectively). Finally, the only significant interaction was Word type x BAS, $F(1, 118) = 4.09$, $p = .046$, $\eta^2_p = 0.033$. Post hoc comparisons with Bonferroni correction showed higher $d'$ values for high-BAS than low-BAS lists for critical lures (0.72 and 0.27, respectively), $t(119) = 4.83$, $p < .001$, 95% CI [0.20, 0.69], $d = 0.38$, but not for studied words (2.42 and 2.23, respectively), $t(119) = 2.03$, $p = .261$, 95% CI [-0.06, 0.43], $d = 0.23$.

Although the participants who performed the task in their L2 knew the meaning of most of the words (86.05%) presented in the experiment, the knowledge of L2-word meaning was not perfect. By contrast, as in the DRM literature, L1 words were very common (e.g., balón, cama, sonrisa, invierno, etc., meaning ball, bed, smile, and winter, respectively), so it is probably safe to assume that undergraduate students knew their meaning. Since it could then be assumed that participants who performed the task in the L2 knew the meaning of fewer words presented in the experiment than participants who performed the task in the L1, one might wonder whether the lower level of false recognition found in the L2, as compared to the L1, could be due to the fact that participants did not know the meaning of all L2 words. To rule out this possibility, we calculated again list-item $d'$ and critical-lure $d'$, but this time only considering the proportion of "yes" responses to L2 words whose meaning was known (see Table 4), leading to the removal of 13.15% of the trials.

We repeated the 2 (word type: studied words, critical lures) x 2 (BAS: high, low) x 2 (language: L1, L2) ANOVA and observed that the results of the previous analysis were replicated. Specifically, this ANOVA showed a significant main effect of word type, $F(1, 118) = 450.55$, $p < .001$, $\eta^2_p = 0.792$, indicating that $d'$ values were higher for the studied words than for the critical lures (2.37 and 0.50, respectively). Moreover, a significant main effect of BAS was found, $F(1, 118) = 21.63$, $p < .001$, $\eta^2_p = 0.155$, with high-BAS lists yielding higher $d'$ values than low-BAS lists (1.59 and 1.27, respectively). A significant main effect of language, $F(1, 118) = 31.45$, $p < .001$, $\eta^2_p = 0.210$, showed that $d'$ values were greater in the L1 than in the L2 (1.68 and 1.18, respectively). Finally, there was a marginally significant interaction Word type x BAS, $F(1, 118) = 3.83$, $p = .053$, $\eta^2_p = 0.031$. As we had an a priori hypothesis regarding this interaction (i.e., BAS will have an effect in false recognition, but not in true recognition), we continue with the analysis of this marginally significant interaction. Post hoc comparisons using Bonferroni correction showed higher $d'$ values for critical lures in high-BAS lists than low-BAS lists (0.72 and 0.28, respectively), $t(119) = 4.71$, $p < .001$, 95% CI [0.19, 0.69], $d = 0.51$, but not for studied words (2.46 and 2.27, respectively), $t(119) = 2.00$, $p = .280$, 95% CI [-0.06, 0.44], $d = 0.22$. Therefore, this analysis confirmed that the differences found in true and false recognition between the L1 and the L2 are not explain by that fact that participants do not have perfect knowledge of L2-word meaning.

**Table 5. Pearson's correlations (*p*-values) between L2 proficiency and *d'* measures for true and false recognition in the L2 as a function of BAS.**

| | Self-reported L2 proficiency |
|---|---|
| Studied words (true recognition) | |
| High BAS | .022 (.433) |
| Low BAS | .097 (.231) |
| High BAS (meaning known) | -.124 (.827) |
| Low BAS (meaning known) | -.118 (.815) |
| Critical lures (false recognition) | |
| High BAS | .228* (.040) |
| Low BAS | .402*** (< .001) |
| High BAS (meaning known) | .249* (.027) |
| Low BAS (meaning known) | .412*** (< .001) |

*$p < .05$
**$p < .01$
***$p < .001$, one-tailed for positive correlations.

In addition, it would be important to examine whether the self-reported L2 proficiency moderates the BAS effect on L2 false memory. Even though to examine the effect of L2 proficiency was not among the main goals of this work, some analyses have been run to test whether the higher the L2 proficiency, the closer the performance to the L1 condition, namely, the higher the false recognition. This would be the expected result based on the RHM as it states that when L2 speakers become more proficient, the conceptual links from L2 words to the concepts become stronger. As shown in Table 5, Pearson's correlations indicated that L2-proficiency was not related to the true recognition of the studied words, but it was positively correlated with the false recognition of the critical lures both from high-BAS and low-BAS DRM lists. That is, the more proficient the participants were in their L2, the greater the false recognition rates of L2 critical lures from both high- and low-BAS lists. Moreover, the results were the same when only the L2 words whose meaning was known were included in the correlation analyses. In other words, irrespective of the knowledge of L2-word meaning, L2-proficiency seem to play an important role in the reported BAS effect on false recognition in the L2.

Finally, regarding the comparison between the L1 and the L2 DRM illusion as a function of BAS, there are two points worth noting. First, in the L1, participants encoded information about the critical lures of both high- and low-BAS lists, being the amount of information greater in the high-BAS than the low-BAS condition (*d'* values of 1.00 vs. 0.59). In contrast, in the L2, even though our last analyses also showed that BAS modulated false recognition, there is a difference with respect to L1 results. Specifically, in the L2, participants only encoded information about the critical lures in the high-BAS condition (*d'* = 0.43) but did not encode information about the critical lures in the low-BAS condition (*d'* = -0.04), as shown in Fig 1. Second, looking into the overall pattern of false recognition, the highest false recognition rate was obtained in the L1 for high-BAS lists, followed by a moderate false recognition rate in the L1 for low-BAS list, but also in the L2 for high-BAS lists. In fact, the false recognition did not differ between L1 low-BAS lists and L2 high-BAS lists, $t(59) = -1.00$, $p = 1.000$, 95% CI [-0.67, 0.30], $d = -0.18$. Lastly, the lowest false recognition rate was obtained in the L2 for low-BAS list.

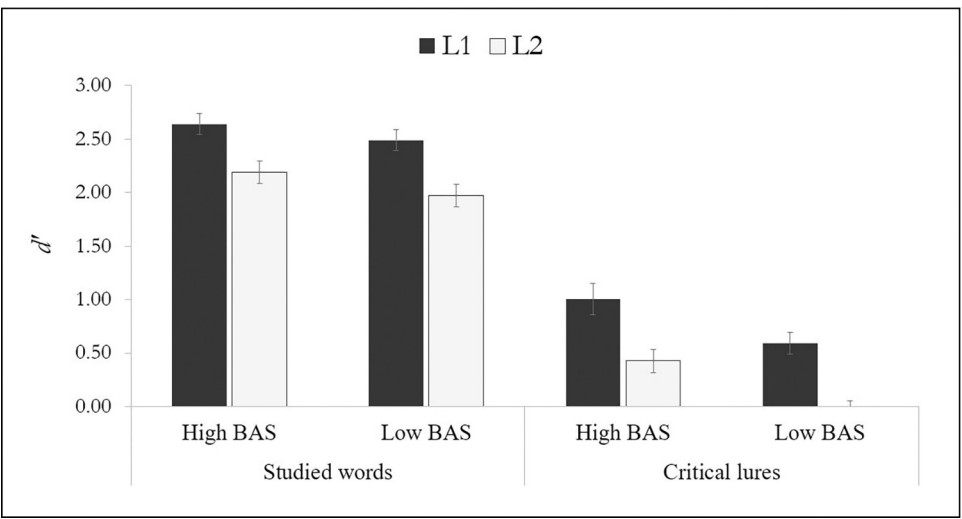

**Fig 1. Values for list-item *d'* and critical-lure *d'* as a function of associative strength (high BAS, low BAS) and language (L1, L2).** The error bars represent the standard error.

## Discussion

The aim of this research was to investigate false memories in the L1 and the L2 as a function of the associative strength between the words of the lists and the critical lures (i.e., backward associative strength or BAS). This study is of interest since the only previous study that, to our knowledge, has investigated this issue had certain limitations that hindered the interpretation of the BAS effect on L2 false recognition [50]. In this sense, we made changes to the experimental materials of our experiment to overcome the limitations of this previous study. This approach is justified given the limited reproducibility of research in the field of psychology [58] and because it helped us clarify and validate Beato and Arndt's [50] findings on the effect of BAS on false memories in the L2. Below, while we discuss the main results, we will highlight the theoretical implications and the novel contributions of the present work.

The most significant improvement and novel contribution raised by our experiment over the prior work by Beato and Arndt [50] was to control the knowledge of L2-word meaning. This was necessary since Beato and Arndt reported that participants had far from perfect knowledge of L2 stimuli and the knowledge of word meaning in their study differed between the high-BAS and low-BAS lists, making it difficult to interpret the BAS effect on false recognition in the L2. To control for the knowledge of L2-word meaning, first, we conducted a previous study to obtain an estimate of the knowledge that L1 speakers (i.e., Spanish) had for the meaning of L2 words (i.e., English). This previous study was useful when building the L2 DRM lists because it allowed us: 1) to include words that were known by most of the participants (participants had near-perfect knowledge of the critical lures, which is vital in order to observe the false memory effect); and 2) to ensure that the knowledge of L2-word meaning was control between high- and low-BAS lists. Second, at the end of the experiment, we asked participants to conduct a translation test, so we had an objective measure of their knowledge regarding the meaning of L2 words. These two approaches provided us with key data to evaluate the effect of BAS in L1 and L2 false recognition ruling out the possibility that the knowledge of L2-word meaning could explain these false memory effects. In sum, when dealing with language processing of second language speakers, it is advisable to control for the knowledge of word meaning when selecting the materials, and to obtain a posterior objective measure of

participants' L2 word meaning knowledge. Although this might seem obvious, in the literature on L2 false memories with the DRM paradigm, until Beato and Arndt's [50] recent study, as far as we know, no one had considered this matter.

The results of our experiment, first, replicated the false recognition effect in the L1, and, moreover, the well-studied effect of BAS on false recognition in this language. That is, false recognition was higher in high-BAS lists than in low-BAS lists [40, 51–54]. This result is explained by the activation-monitoring framework or AMF [40] by proposing that, the higher the associative strength from the studied words to the critical lure (i.e., BAS), the greater the probability that the critical lure will be falsely recognized as a studied words at the memory test. Second, regarding the false memory effect in the L2 as a function of BAS, as expected, we found false recognition in the L2 or the non-dominant language [50, 60–65] (see also [39] for a review). Moreover, our data showed that the false recognition in the L2 was modulated by BAS. That is, L2 false recognition was higher when the association between the words in the list and the critical lures was high (i.e., high-BAS lists) than when it was low (i.e., low-BAS lists). This BAS effect in the L2 replicated the findings of Beato and Arndt [50], but this time using ideal experimental conditions for the study of the effect of BAS in isolation in the L2: 1) High- and low-BAS lists that did not differ in terms of the knowledge of L2-word meaning, 2) DRM lists had only one critical lure, rather than three critical lures per lists as used in the previous study, and 3) The DRM lists had FAS values of virtually zero. In addition, the ANOVAs performed only with the words whose meaning was known to the participants clarifies the possible effect that the lack of knowledge of the L2 words could have on our results and conclusions, providing the scientific literature with solid evidence that supports the RHM and the AMF.

Following the signal detection theory [78], the critical-lure $d'$ values in the L2 indicated that the difference in this language in false recognition between the high- and low-BAS lists was due to the fact that participants had not encoded information about the critical lures in the low-BAS condition ($d'$ value almost equal to zero) but had encoded information about the critical lures in the high-BAS condition ($d' = 0.43$). That is, it seems that critical lures in low-BAS lists were not activated during the study phase, in contrast to the critical lures in the high-BAS lists which were activated. In other words, our data showed that, in the L2, the spreading activation only reached concepts that were strongly associated with the words in the list but did not extend to concepts that were further away in the semantic memory. These findings go in line with the theoretical models focusing on the concept of fuzziness as the main feature of the L2 lexicon [46, 56]. These models propose that semantic-level representations of L2 words, as compared to L1 words, have fewer connections and these connections tend to be weaker. Accordingly, it would be reasonable to think that the connections in the L2 lexicon are mainly between words that are highly associated (i.e., high-BAS lists), but not between words that are further apart in this lexicon (i.e., low-BAS lists). Therefore, this could explain why the false recognition effect in the L2 was found only when the associative strength between the words in the list and the critical lures was high. Moreover, our data also showed evidence that the self-reported L2 proficiency plays an important role in the L2 false memory effect. Specifically, we found that the higher the L2 proficiency, the greater the false recognition of L2 critical lures from high-BAS and low-BAS lists. These results are consistent with previous studies [39, 50] and support the predictions of both the RHM and the AMF. The RHM predicts that once L2 speakers become more proficient, their conceptual links from L2 lexical representations strengthened, activating the conceptual representation from L2 words faster and more directly as their proficiency in this language increases. Following the AMF, this predicts a greater production of false memories as L2 proficiency improves.

Finally, we were interested in examining how concepts are activated from words in the L1 and the L2, and how that activation spreads throughout the semantic memory. To this aim, we

compared the false recognition effect between the L1 and the L2 as a function of BAS. In line with our hypothesis, the results replicated the findings of previous studies showing that false recognition was higher in the L1 than in the L2 [50, 61, 62, 65], and as in Beato and Arndt's [50] study, this finding was extended not only to high-BAS lists, but also to low-BAS lists. Furthermore, using the sensitivity index (*d'*) from the signal detection theory, our experiment has made a contribution of great interest to the literature, since, although the overall false recognition was found in both languages and was modulated by BAS, the critical-lure *d'* values allowed us to observe a qualitative difference between the L1 and the L2. Precisely, false recognition (as measured by the sensitivity index) was found in both high- and low-BAS lists in the L1. However, in the L2, the overall decrease in false recognition in this language led the effect to appear only in high-BAS lists, but not in low BAS-lists (see Fig 1).

The higher false recognition in the L1 or dominant language than in the L2 or non-dominant language can be explained by two theories from different research fields, whose explanations complement each other to account for our results: the activation-monitoring framework (AMF) [40] and the revised hierarchical model (RHM) [41, 42]. According to the AMF, when participants study lists of words, not only conceptual representations of those words are activated, but also associated concepts, such as the critical lures. Moreover, in line with the RHM, conceptual representations are activated more directly and automatically from words in the L1 than in the L2. Therefore, considering the predictions of both theories, conceptual representations of critical lures will be more activated when the lists of words were studied in the L1 than in the L2, leading to a higher false recognition in the former than in the latter language, irrespective of BAS.

Looking at the overall pattern of levels of activation throughout the L1 and L2 (see Fig 1), we reported an interesting pattern of results. The L1 critical lures form high-BAS lists were the ones that received the highest activation in our experiment, followed by the moderate activation received by the L1 critical lures from low-BAS lists and the L2 critical lures from high-BAS lists, which did not differ between them. Finally, the L2 critical lures from low-BAS lists, as it was previously mentioned, barely received any activation. This seems to indicate the importance of critical lure activation that would be expected under the AMF. This pattern of results could be explained by the ontogenesis model of L2 lexical representation [46, 56] as it states that semantic-level representations of L2 words, as compared to L1 words, have fewer connections and, especially important for our data, these connections tend to be weaker. With this in mind, we can infer from our results that strongly associated words in the L2 would be associated in second-language learners with a similar level of associative strength as weakly associated words in their L1.

In summary, participants who performed the task in their L1 activated the conceptual representations of critical lures from both high- and low-BAS lists, reflecting the rapid access to word meaning in the L1 and the existence of numerous connections between the concepts associated with these words, which made it possible to automatically activate even words that are further away in the lexicon (i.e., critical words from low-BAS lists). Besides, as expected, critical lures from high-BAS lists received more activation than critical words from low-BAS lists, demonstrating that activation spread in a decreasing gradient. For its part, participants who performed the task in their L2 showed activation only for critical lures that were strongly associated with the words in the list, but not for those words that were further away in the lexicon. Moreover, the activation of these critical lures in the L2 was, in general, lower than that of critical lures in the L1. This reflects a more diffuse and imprecise network in the L2 [46, 56, 57], compared to the L1, with weaker connections between the word forms and their conceptual representations [41].

We would like to highlight some limitations, suggestions for future research, and practical implications. First, although the subjective (i.e., self-report) and objective (i.e., translation test score) L2-proficiency measures used in this experiment were positively correlated ($r = .74$, $p <$ .001), future research might benefit from using standardized questionnaires to assess language proficiency [85–87]. In addition, following the reviewer's suggestion, it would have been advisable to also collect the translation test data for L2 words in participants tested in their L1. This would be useful to know if both groups of participants had similar vocabulary knowledge in their L2. Second, it would be interesting to examine whether L2 proficiency moderates the results reported in the present experiment using participants with a wider range of L2 proficiency. Third, one should bear in mind that the false memory effect in the DRM paradigm is not only affected by the activation processes mentioned above, but other monitoring processes may also be involved. Therefore, to be able to isolate the automatic processes that reflect the way in which lexical and conceptual representations are accessed from L1 and L2 words, future research could use experimental procedures that study the activation processes more directly, such as, for example, the semantic priming paradigm. Fourth, cross-language effects should be also addressed in future studies to further assess the theoretical frameworks (revised hierarchical model and activation monitoring framework) [65, 88]. Lastly, our findings on how the L2 lexicon is organized and how conceptual representations are accessed from L2 words may have practical implications for the second language learning field.

As a final remark, this study makes the following contributions to the scientific literature: 1) It highlights the importance of assessing the knowledge of word meaning in a second language when studying effects involving access to conceptual representations (e.g., false memory effect or semantic priming effect); 2) Our data confirmed that the procedure used in this experiment for the construction of the L2 DRM lists succeeded in correctly estimating and controlling for the knowledge of the meaning of L2 words; 3) For the first time in the literature, the effect of BAS, one of the most studied variables in the DRM literature, has been investigated on false recognition in the L2 by controlling for the knowledge of L2-word meaning; and 4) By studying the associative memory illusion in the L1 and the L2 as a function of BAS, we have been able to provide evidence for the scarcity and weakness of connections in the L2 lexicon, as L2 false recognition occurred when the association between concepts was strong (i.e., high-BAS lists), but not when it was weak (i.e., low-BAS lists).

## Author Contributions

**Conceptualization:** Mar Suarez, Maria Soledad Beato.

**Data curation:** Mar Suarez, Maria Soledad Beato.

**Formal analysis:** Mar Suarez.

**Funding acquisition:** Mar Suarez.

**Investigation:** Maria Soledad Beato.

**Methodology:** Mar Suarez, Maria Soledad Beato.

**Supervision:** Maria Soledad Beato.

**Validation:** Maria Soledad Beato.

**Writing – original draft:** Mar Suarez.

**Writing – review & editing:** Mar Suarez, Maria Soledad Beato.

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
