## [Decision Letter · Decision Letter 0]

12 Dec 2022

PONE-D-22-27134False memory in a second language: the importance of controlling the knowledge of word meaningPLOS ONE

Dear Dr. Suarez,

Thank you for submitting your manuscript to PLOS ONE. After careful consideration, we feel that it has merit but does not fully meet PLOS ONE’s publication criteria as it currently stands. Therefore, we invite you to submit a revised version of the manuscript that addresses the points raised during the review process.

As you can see below, three expert reviewers found your research topic very relevant and acknowledge the effort you put into conducting and writing this research.

They also offer extensive and extremely valuable comments to improve your manuscript. However, while the reviewers have identified some merits, there are also conceptual and methodological issues that should be fully addressed. Overall, from my own assessment, I agree with most of the presented comments. I am not going to reiterate them all. Still, I would suggest particular attention to the following:

While replication studies are relevant, and the authors even identified and addressed some methodological issues from Beato and Ardnt’s study, I concur with Reviewer 1 regarding the lack of theoretical advance or novel contribution. In the revised version, the authors should emphasize in the introduction and the discussion how this replication and the methodological issues you are addressing might contribute to informing and advancing theory.Reviewer 2 suggests addressing the fuzzy-trace theory. I suggest at least some mention of explanations about L1_L2 processing differences (e.g., cognitive, affective) that might apply to these results.Some methodological choices should also be justified. Why did the authors use a between design instead of comparing L1 and L2 performance within the same participants (Reviewer 2)? Why did the authors use a self-reported measure of L2 proficiency instead of a standardized test (Reviewer 1)? Is the self-reported score significantly above average? Ideally, given that performance in L2 might be related to several other variables, all participants should pass the test and then be randomly assigned to one of the conditions.While (as the authors did) it is important to control for knowledge of the words' meaning (and repeating the analysis including only the words that were known), it would be important to examine whether L2 proficiency (even if self-reported) moderates the results (Reviewer 3). It would be expected that the higher the proficiency, the closer the performance to the L1 condition?. This analysis could also be implemented using the “translation” indicators.As long as there is an a priori hypothesis regarding the specific comparison, I have no problems with the post-hoc test following a non-significant interaction (Reviewer 3). Still, this needs to be better justified.All the remaining comments of the reviewers should be comprehensively addressed.

We look forward to receiving your revised manuscript.

Kind regards,

Margarida Vaz Garrido

Academic Editor

PLOS ONE

Journal Requirements:

Additional Editor Comments:

As you can see below, three expert reviewers found your research topic very relevant and acknowledge the effort you put into conducting and writing this research.

They also offer extensive and extremely valuable comments to improve your manuscript. However, while the reviewers have identified some merits, there are also conceptual and methodological issues that should be fully addressed. Overall, from my own assessment, I agree with most of the presented comments. I am not going to reiterate them all. Still, I would suggest particular attention to the following:

1. While replication studies are relevant, and the authors even identified and addressed some methodological issues from Beato and Ardnt’s study, I concur with Reviewer 1 regarding the lack of theoretical advance or novel contribution. In the revised version, the authors should emphasize in the introduction and the discussion how this replication and the methodological issues you are addressing might contribute to informing and advancing theory.

2. Reviewer 2 suggests addressing the fuzzy-trace theory. I suggest at least some mention of explanations about L1_L2 processing differences (e.g., cognitive, affective) that might apply to these results.

3. Some methodological choices should also be justified. Why did the authors use a between design instead of comparing L1 and L2 performance within the same participants (Reviewer 2)? Why did the authors use a self-reported measure of L2 proficiency instead of a standardized test (Reviewer 1)? Is the self-reported score significantly above average? Ideally, given that performance in L2 might be related to several other variables, all participants should pass the test and then be randomly assigned to one of the conditions.

4. While (as the authors did) it is important to control for knowledge of the words' meaning (and repeating the analysis including only the words that were known), it would be important to examine whether L2 proficiency (even if self-reported) moderates the results (Reviewer 3). It would be expected that the higher the proficiency, the closer the performance to the L1 condition?. This analysis could also be implemented using the “translation” indicators.

5. As long as there is an a priori hypothesis regarding the specific comparison, I have no problems with the post-hoc test following a non-significant interaction (Reviewer 3). Still, this needs to be better justified.

6. All the remaining comments of the reviewers should be comprehensively addressed.

Reviewers' comments:

Reviewer's Responses to Questions

**Comments to the Author**

1. Is the manuscript technically sound, and do the data support the conclusions?

Reviewer #1: Yes

Reviewer #2: Yes

Reviewer #3: Yes

2. Has the statistical analysis been performed appropriately and rigorously? 

Reviewer #1: Yes

Reviewer #2: Yes

Reviewer #3: No

3. Have the authors made all data underlying the findings in their manuscript fully available?

Reviewer #1: Yes

Reviewer #2: Yes

Reviewer #3: Yes

4. Is the manuscript presented in an intelligible fashion and written in standard English?

Reviewer #1: Yes

Reviewer #2: Yes

Reviewer #3: Yes

5. Review Comments to the Author

Reviewer #1: This paper presents a single experiment focused on the study of false memories in second language with use of the DRM paradigm. The experiment builds upon Beato and Ardnt’ earliest study with the main aim of overcoming their limitations.

Despite the replication of a single study with the implementation of methodological improvements is always desirable and welcome, the current work lacks of any novelty either by the inclusion of a new variable or a follow-up study extending the mere replication of the results. Therefore, there is a lack of theoretical advance or novel contribution. Moreover, it has some issues with the order of presentation of the information in both, introduction and discussion.

Yet, I acknowledge that the topic of research is currently of utmost relevance, the effort and scrupulosity in the methods and result section.

Thus, the current research I do not consider it complies with PLOSone standards for publication unless a new follow-up study and major changes in the text are included.

Here there are several comments the authors might consider in future submissions:

In page 6, starting in line 124, three main issues are raised from Beato and Ardnt:

a- the use of 3 lures instead of 1

b- DRM list varied in levels of BAS and FAS

c- Not control of participants’ meaning knowledge for L2 words

In page 7, starting in line 146, authors presented what they implemented to overcome these three

issues but in a different order (c, a, X, b) which disturb the readiness of the manuscript.

In the discussion it is again found this lack of order. Including such detailed order of hypotheses and objectives in the introduction should have prompt to follow it in the discussion.

Moreover, the presented study itself also presents some limitations that should have been the least mentioned there: for example, instead of self-report on second language command, an standardize test could have been used.

Discussion:

Page 24, line 497:

“Although this might seem obvious, in the literature on L2 false memories, until Beato and Arndt’s [44] recent study, as far as we know, no one had considered this matter”.

Change for:

“Although this might seem obvious, in the literature on L2 false memories with the use of DRM, until Beato and Arndt’s [44] recent study, as far as we know, no one had considered this matter”.

Page 27, line 582:

I would suggest to re-write this sentence:

“As a final remark, we would like to suggest some practical implications and future research”.

Because what it follows it is only suggested one practical thing: “[…] future research may benefit from employing experimental procedures that study the activation processes more directly, such as, for example, the semantic priming paradigm”.

Reviewer #2: This study is investigating false memories using the DRM paradigm in both L1 and L2 for bilingual individuals. Unlike previous studies, this study examined the importance of using both high and low BAS lists but made sure that the lists were equivalent in participants’ knowledge of the meaning of words in the lists. Even when controlling for knowledge of word meaning, the authors showed higher false recognition with high-BAS lists than in low-BAS lists.

Overall, I thought the authors did a very nice job of setting up the study, guiding the reader through the logic for the study, and explaining the need for the study. I also thought their methodology and attention to detail in setting up their study was also excellent.

While I think the authors did an excellent job of discussing and introducing the two theories (AMT and RHM), which I believe to be very relevant to their study, some individuals might like to see some mention of the fuzzy-trace theory. I think the AMT and RHM theories are enough, but the fuzzy-trace theory is the other primary theory in the false memory literature.

On page 9 (in the Participants section) and page 13 (in the Procedure section), the authors mentioned that participants were only tested in either their L1 language (Spanish) or their L2 language (English). I was curious why this was done. Why not test participants in both their L1 and their L2 language?

On page 14: The authors stated the participants who were tested in their L2 were asked to translate the 100 English words that they had been presented with during the study. However, on page 21 (lines 449-451) the authors mentioned that d’ scores were calculated for the proportion of “yes” responses to L2 words whose meaning was known. Initially, it wasn’t clear to me if the participants provided the meaning for each of the words or if they were simply asked to indicate if they knew the word meaning (a yes or no response). While I did understand what the authors were meaning here, it did take me a few different readings to fully understand what they meant. I’m not sure if this was an issue with the authors explanation of this or me being somewhat dense in not understanding it quickly. I actually would have liked to see both participants who were tested in their L1 language and those who were tested in their L2 language complete this task, so that a comparison could have been made on their general knowledge of English words. Obviously this was more important to do for those tested in their L2, so I understand why the authors did this task as they did. A related issue is on page 15 where the authors mention that participants were most successful in knowing the meaning of the critical lures compared to any other word types (i.e., list items, distractors, etc.). I would expect that this is an important necessity for participants to recognize the critical lures and could be an alternative factor that would help explain differences in L1 and L2 lists. This could be indicative of a gist-based representation that is consistent with fuzzy-trace theory.

On page 16 (lines 330-334): The authors mentioned that there was a Word Type x BAS interaction. However, in the comparison is seems unlikely as studied words and critical lure recognition seemed to be nearly identical for both the high-BAS (.83 and .98, respectively) and low-BAS lists (.84 and .97, respectively). Perhaps I’m missing something, but this doesn’t make sense. Maybe adding a table of means for the L2 Knowledge of word meaning would be helpful to see these data more clearly. In this table the authors could provide the percent/proportion of each word type that was known by participants.

On page 19 (lines 390-392): The authors mentioned that the L2 English DRM lists created in their study produced false memories. However, it seems apparent that they did not produce false memories to the same extent as they did in the L1. This might be worth noting here.

On page 21 (lines 445-448): The authors mentioned that participants’ who were tested in the L2 knew the meaning of fewer word presented than participants who performed the task in the L1. While this statement is probably true, I didn’t think the authors tested participants for the meaning of words in the L1. I thought only L2 participants were tested on their knowledge of the meaning of words. On page 22 (lines 464-466), the authors stated that their analysis confirmed that the differences in true and false recognition between the L1 and L2 participants was not due to a lower knowledge of L2-word meaning. Again, if only the L2 participants were tested on their knowledge of the word meanings, this seems somewhat confusing.

I thought the authors’ explanation for why participants showed more false memories for the L2 word lists was higher with the high-BAS lists than the low-BAS lists on page 25 was very clear and organized. It appears that they authors found a nice pattern of results such that false memory was highest in L1 for high-BAS lists, moderate in L1 for low-BAS lists and also moderate for L2 high-BAS lists, and lowest for L2 for low-BAS lists. These seem to indicate the importance of critical lure activation that would be expected for the AMT. The only shortcoming that I see is that I don’t think the authors made the distinction between the cognitive distinction between the L1/low-BAS lists and the L2/high-BAS lists. What do the authors think is going on between these two conditions and how might they be explained with the AMT, RHM, or Fuzzy-trace theories?

Overall, I enjoyed the article and thought the authors did an excellent job at conducting, presenting, and explaining their research. Below I have a few small wording changes to help the authors.

Wording Suggestions

Line 44: change wording – change from “has been favored by the fact” to “has been influenced by the fact”

Line 83: change wording – change from “the critical lures has” to “the critical lures have”

Line 161: change wording – change “cero” to “zero”

Line 168: change wording – change “Arndt” to “Arndt’s”

Line 190: change wording – change “when they had” to “when they were”

Line 365: change wording – change “Once we have proofed” to “Once we have proved”

Line 515: change wording – change “cero” to “zero”

Reviewer #3: PONE-D-22-27134

False memory in a second language:

The importance of controlling the knowledge of word meaning

Summary:

This research examined true and false recognition using DRM lists in participants’ native language Spanish (L1) or in participants’ second language (L2). The authors use activation monitoring framework and the revised hierarchical model to deduce their hypotheses and predictions. Participants viewed 16 lists with six studied words per list, in either L1 or L2. Within each language group, ½ the lists had a high backward associative strength (BAS) and half had a low BAS. Across languages, researchers controlled for BAS values, forward associative strength (FAS) values, and L2 knowledge of word meaning. Additionally, all participants self-reported a moderate level of fluency and there appeared to be no differences across sociodemographic variables. The results supported the authors’ hypotheses of greater false recognition in L1 than L2 as well as higher false recognition for high BAS vs low BAS DRM lists; moreover, the false recognition effect in L2 was only found for high BAS DRM lists. The results replicate findings reported by Beato and Arndt (2021), but used a between-subject design, different materials, and removed a potential confound of L2 knowledge of word meaning. Though the authors do demonstrate a more tightly controlled experiment (with respect to BAS) and support their analyses using d’ analyses, they replicate previous findings concerning the modulating effect of BAS on false recognition in L2.

Main Points:

1. One question I had was why the authors’ chose to use different DRM lists for their high and low BAS conditions? In Beato and Arndt’s work, they kept the critical lure constant, using top four and bottom four associates of the same critical lure to create the high and low BAS conditions, respectively. This was done to ensure that there were any item differences that could occur with using different lists. Though BAS values did not differ across DRM list language, can the authors be sure there are no item differences? In addition, why did the authors manipulate DRM list language as a between-subject variable rather than a within-subject variable?

2. Another question I had is about fluency. Though the authors report that participants self-reported as moderately fluent (M = 6.00) and that there were no differences between participants across each list language group, the authors do not mention predictions made by the revised hierarchical model with respect to fluency. In their E2, Beato and Arndt find that the effects of BAS on false recognition in L2 are also influenced by differences in participants’ fluency. It is expected that as participants become more fluent in their L2 that the links between L2 and concepts strengthen, which also has implications for activation monitoring framework. Thus, it appears that irrespective of L2 word knowledge, fluency may play an important role in the reported BAS effects on false recognition in L2.

3. I guess I am also curious about cross-language effects (L1 studied list to L2 test, vice versa), as I think it would really test the revised hierarchical model and AMF as well as the effects of BAS.

4. On page 19, the authors perform a repeated measure ANOVA and indicate they used a Bonferroni correction for post hoc tests. The authors report a significant pairwise comparison finding a higher proportion “yes” responses to critical lures than to distractors. The exact p-value reported is .038. Is this value significant when applying the Bonferroni correction for experimentwise error? How many tests are being performed here?

5. Similarly, on page 22, the authors report a “marginally significant interaction with Word type x BAS). Given the reported p-value is .053, this value is not significant if using a .05 significance value. In my opinion, the authors should not have proceeded with conducting post-hoc analyses on their data.

Minor typo. Reference #57, Diliberto(-Macaluso) is misspelled as Deliberto.

6. PLOS authors have the option to publish the peer review history of their article (what does this mean?). If published, this will include your full peer review and any attached files.

Reviewer #1: No

Reviewer #2: **Yes: **Jeffrey S. Anastasi

Reviewer #3: **Yes: **Kristen Diliberto-Macaluso

---

## [Author Response · Author response to Decision Letter 0]

25 Jan 2023

January 24th, 2023 

Dear Dr. Garrido:

Thank you very much for your response. We are very grateful for the suggestions you and the reviewers made on the article. 

In the first place we would like to state how much we appreciated the very detailed set of comments from both the academic editor and the reviewers to the paper PONE-D-22-27134 entitled “False memory in a second language: the importance of controlling the knowledge of word meaning”. Undoubtedly, all your comments were very useful to improve the manuscript, making now a much stronger contribution to the literature. 

The paper has been modified to clarify some questions and to respond to the comments. As requested, we are sending three documents: 1) a rebuttal letter that responds to each point raised by the academic editor and reviewer (please, see Responde to Reviewers.pdf); 2) a marked-up copy of the manuscript that highlights the changes made to the original version (please, see Revised Manuscript with Track Changes.doc); and 3) an unmarked version of the revised paper without tracked changes (please, see Manuscript.doc).

We hope these changes answer the reviewers and the editor and explain the changes that have been made in the paper.

I look forward to hearing from you. 

Yours sincerely,

Mar Suarez

University of Salamanca

Faculty of Psychology

Department of Basic Psychology, Psychobiology, and Methodology of Behavioural Sciences

Avda. de la Merced, 109-131

37005 Salamanca, Spain

E-mail: marsuarez@usal.es

---

## [Decision Letter · Decision Letter 1]

11 Apr 2023

PONE-D-22-27134R1False memory in a second language: the importance of controlling the knowledge of word meaningPLOS ONE

Dear Dr. Suarez,

Thank you for submitting your manuscript to PLOS ONE. After careful consideration, we feel that it has merit but does not fully meet PLOS ONE’s publication criteria as it currently stands. Therefore, we invite you to submit a revised version of the manuscript that addresses the points raised during the review process.

We look forward to receiving your revised manuscript.

Kind regards,

Margarida Vaz Garrido

Academic Editor

PLOS ONE

Journal Requirements:

Additional Editor Comments:

First of all, I apologize for the length of the reviewing process. Unfortunately, one of the previous reviewers could not revise the paper again, and the two remaining reviewers presented quite different views regarding the resubmitted version. One reviewer recommended accepting the paper as it is, and the other recommended not accepting it on the grounds of lack of novelty to the current state of the art. While I agree that theoretical innovation is important, novelty is not a mandatory requirement to publish in PLOS ONE. I also acknowledge the authors' efforts in detailing how the current replication overcomes the limitations of previous work. Still, I invited an additional expert that also concurs with this view. I am not recommending accepting the paper as it is, but I am confident that the authors will be able to address the issues raised by reviewer 4. I also recommend the authors to provide a point-by-point reply to all the comments raised by this reviewer, including an indication of the page/line where they introduced changes.

Reviewers' comments:

Reviewer's Responses to Questions

**Comments to the Author**

1. If the authors have adequately addressed your comments raised in a previous round of review and you feel that this manuscript is now acceptable for publication, you may indicate that here to bypass the “Comments to the Author” section, enter your conflict of interest statement in the “Confidential to Editor” section, and submit your "Accept" recommendation.

Reviewer #1: (No Response)

Reviewer #3: All comments have been addressed

Reviewer #4: (No Response)

2. Is the manuscript technically sound, and do the data support the conclusions?

Reviewer #1: Partly

Reviewer #3: Yes

Reviewer #4: Yes

3. Has the statistical analysis been performed appropriately and rigorously? 

Reviewer #1: Yes

Reviewer #3: Yes

Reviewer #4: Yes

4. Have the authors made all data underlying the findings in their manuscript fully available?

Reviewer #1: Yes

Reviewer #3: Yes

Reviewer #4: Yes

5. Is the manuscript presented in an intelligible fashion and written in standard English?

Reviewer #1: Yes

Reviewer #3: Yes

Reviewer #4: Yes

6. Review Comments to the Author

Reviewer #1: I am revising this article for second time, and unfortunately I have to recommed to Reject it.

As mentioned in my previous review, in order to reach PLOSone standards for publication this manuscript requires a follow-up study which is not included in the new version. Withouth this follow-up study the manuscript lacks of novelty to the current state of art.

Finally, the letter with the detailed answer for each Reviewers´comments is just a mere general cover letter. For future submissions I would suggest to the authors to provide a point by point answer to all Reviewers´comments including page/line where the changes were done.

Reviewer #3: I have re-read the manuscript including the changes and additions the authors made to the manuscript. I believe that they have sufficiently addressed each of the reviewer's questions in their revisions. The authors ran a carefully controlled study ruling out the potential influence of a 3rd variable, L2 word knowledge, on false recognition of high and low BAS DRM lists (while controlling for FAS across lists). The work supports existing theories of false memory and second language acquisition. I recommend the paper for publication.

Reviewer #4: The manuscript entitled "False memory in a second language: the importance of controlling the knowledge of word meaning" aims to study the effect of the backward associative strength in false memories in native and non-native languages. It is a replication study that overcomes the limitations of a previous one and thus allows the authors to make stronger inferences regarding the BAS effect in false memories in L1 and L2.

I have received the revision of this work and reviewed all previous reviewers' comments and the authors' responses. I do recommend the publication of this version. I find that they have successfully improved the former manuscript following the recommendations of the reviewers and the editor. I enjoyed reading this work, and I believe it is well-motivated, with a test of important theoretical approaches in both the field of false memory and bilingualism. It is well-written, and the methods are rigorous, with exquisite control of the materials used. Also, I appreciate that they made available materials and data.

Below I have added a couple of comments.

Comments:

Page 7, line 145. In the introduction, I recommend succinctly explaining, with a couple of words, some important concepts for potential readers that might not be familiar with them. For example, this research might interest psycholinguists unfamiliar with BAS and FAS or the main idea of the fuzzy-trace theory.

In the methods, did the authors control for frequency in the lists/critical lures? Word frequency is important when dealing with concept retrieval in activation accounts. As a reader, I would like to know whether it has impacted the BAS manipulation (i.e., the high-BAS words could potentially present greater frequency than low-BAS and differences within each language).

Finally, a couple of typos:

Page 28, line 627. In "These results are consistent with previous studied", it should be "studies".

Page 8, line 166. "the BAS effect occur" should be "occurs".

7. PLOS authors have the option to publish the peer review history of their article (what does this mean?). If published, this will include your full peer review and any attached files.

Reviewer #1: No

Reviewer #3: **Yes: **Kristen Diliberto-Macaluso

Reviewer #4: No

---

## [Author Response · Author response to Decision Letter 1]

18 Apr 2023

JOURNAL REQUIREMENTS:

RESPONSE: The authors believe that the reference list is complete and correct, and that they have not cited any retracted paper. We corrected a typo in the journal name of reference 59, and included three new references that were needed to properly implement the changes suggested by reviewer 4. The new references are the followings: 

49. Beato MS, Suarez M, Cadavid S. Disentangling the effects of backward/forward associative strength and theme identifiability in false memory. Psicothema. 2023;35: 178–188. doi:10.7334/psicothema2022.288

69. Duchon A, Perea M, Sebastián-Gallés N, Martí A, Carreiras M. EsPal: one-stop shopping for Spanish word properties. Behav Res Methods. 2013;45: 1246–1258. doi:10.3758/s13428-013-0326-1

70. Brysbaert M, New B. Moving beyond Kučera and Francis: a critical evaluation of current word frequency norms and the introduction of a new and improved word frequency measure for American English. Behav Res Methods. 2009;41: 977–990. doi:10.3758/BRM.41.4.977

REVIEWER #4

REVIEWER COMMENT: I have received the revision of this work and reviewed all previous reviewers' comments and the authors' responses. I do recommend the publication of this version. I find that they have successfully improved the former manuscript following the recommendations of the reviewers and the editor. I enjoyed reading this work, and I believe it is well-motivated, with a test of important theoretical approaches in both the field of false memory and bilingualism. It is well-written, and the methods are rigorous, with exquisite control of the materials used. Also, I appreciate that they made available materials and data.

RESPONSE: We thank Reviewer 4 for the kind words and the feedback on our manuscript. All the thoughtful comments on our submission have been considered and implemented. We believe that the manuscript is stronger and clearer now.

REVIEWER COMMENT: Page 7, line 145. In the introduction, I recommend succinctly explaining, with a couple of words, some important concepts for potential readers that might not be familiar with them. For example, this research might interest psycholinguists unfamiliar with BAS and FAS or the main idea of the fuzzy-trace theory.

RESPONSE: Thank you for raising this important topic. As suggested, we have explained, with a couple of words, the concepts of BAS and FAS, and the main idea of the fuzzy-trace theory. We have also referred the reader to useful literature about these topics. We think that having included these brief explanations will be useful to some readers. The text included in the manuscript is as follows:

[Page 6/lines 130-131] “Briefly, BAS refers to the associative strength from the studied items to the critical lure [21,22,49].”

[Page 7/lines 148-150] “Finally, Beato and Arndt used DRM lists that varied in the levels of both backward (BAS) and forward associative strength (FAS or the associative strength from the critical lure to the studied items; see [49] for a recent study independently examining the effect of BAS and FAS on false memory).”

[Page 4/lines 77-79] “An alternative explanation for the false memory effect is given by the fuzzy-trace theory, which states that false memory appears when the gist information of the list (matching the critical lure) is extracted, and the retrieval of verbatim representations is not enough to reject the critical lure (see [43,44] for more details).”

REVIEWER COMMENT: In the methods, did the authors control for frequency in the lists/critical lures? Word frequency is important when dealing with concept retrieval in activation accounts. As a reader, I would like to know whether it has impacted the BAS manipulation (i.e., the high-BAS words could potentially present greater frequency than low-BAS and differences within each language).

RESPONSE: We thank the reviewer for bringing this matter up. Yes, we control for frequency (and word length) in high-BAS vs. low-BAS lists in both languages. We agree with the reviewer that frequency is an important variable when dealing with lexical access and concept retrieval. We have made that information available for the reader in the method section: 

[Page 13/lines 264-268] “Furthermore, words from the two BAS conditions did not differ in terms of word length (i.e., number of letters), t(14) = 1.78, p = .097, 95% CI [-0.10, 1.06], d = 0.89, nor in their frequency (i.e., logarithmic frequency from EsPal [69]), t(14) = 0.62, p = .542, 95% CI [-0.22, 0.40], d = 0.31, which was, on average, 1.13 (SD = 0.30) for high-BAS lists and 1.22 (SD = 0.27) for low-BAS lists.”

[Page 14/lines 291-294] “At the same time, high- and low-BAS L2 lists did not differ in their FAS values, t(14) = -0.85, p = .409, 95% CI [-0.01, 0.01], d = -0.43, word length, t(14) = 0.29, p = .779, 95% CI [-0.59, 0.77], d = 0.14, or frequency (i.e., logarithmic frequency based on SUBTLEXUS [70]), t(14) = 0.48, p = .641, 95% CI [-0.19, 0.30], d = 0.24, with frequency mean values of 1.61 (SD = 0.22) and 1.66 (SD = 0.24) for high- and low-BAS lists, respectively.”

REVIEWER COMMENT: Finally, a couple of typos:

Page 28, line 627. In "These results are consistent with previous studied", it should be "studies".

Page 8, line 166. "the BAS effect occur" should be "occurs".

RESPONSE: Thank you very much for your suggestions. We have corrected the typos in the revised manuscript. The new text now appears in page 8 (lines 169-170) and page 30 (lines 638).

---

## [Editor Report · Decision Letter 2]

2 May 2023

False memory in a second language: the importance of controlling the knowledge of word meaning

PONE-D-22-27134R2

Dear Dr. Suarez

We’re pleased to inform you that your manuscript has been judged scientifically suitable for publication and will be formally accepted for publication once it meets all outstanding technical requirements.

Kind regards,

Margarida Vaz Garrido

Academic Editor

PLOS ONE

Additional Editor Comments (optional):

Thank you for addressing all the comments so thoroughly and reviewing the manuscript accordingly. I think the final result constitutes an interesting contribution to field.

---

## [Editor Report · Acceptance letter]

3 May 2023

PONE-D-22-27134R2 

False memory in a second language: the importance of controlling the knowledge of word meaning 

Dear Dr. Suarez:

I'm pleased to inform you that your manuscript has been deemed suitable for publication in PLOS ONE. Congratulations! Your manuscript is now with our production department. 

Kind regards, 

on behalf of

Dr. Margarida Vaz Garrido 

Academic Editor

PLOS ONE